# Quasi-experimental evaluation of national border closures on COVID-19 transmission

**Mathieu J. P. Poirier** [1,2], **Susan Rogers Van Katwyk** [1], **Gigi Lin** [1], **Steven J. Hoffman** [1,2,3]*

**1** Global Strategy Lab, Dahdaleh Institute for Global Health Research, York University, Toronto, Canada, **2** School of Global Health, Faculty of Health, York University, Toronto, Canada, **3** Department of Health Research Methods, Evidence, and Impact and McMaster Health Forum, McMaster University, Hamilton, Canada

* steven.hoffman@globalstrategylab.org

**Data Availability Statement:** All data, code, and results for analyses are either publicly available, as referenced in Supplementary Materials, or made available through Scholars Portal Dataverse at

## Abstract

With over 200 pandemic threats emerging every year, the efficacy of closing national borders to control the transmission of disease in the first months of a pandemic remains a critically important question. Previous studies offer conflicting evidence for the potential effects of these closures on COVID-19 transmission and no study has yet empirically evaluated the global impact of border closures using quasi-experimental methods and real-world data. We triangulate results from interrupted time-series analysis, meta-regression, coarsened exact matching, and an extensive series of robustness checks to evaluate the effect of 166 countries' national border closures on the global transmission of COVID-19. Total border closures banning non-essential travel from all countries and (to a lesser extent) targeted border closures banning travel from specific countries had some effect on temporarily slowing COVID-19 transmission in those countries that implemented them. In contrast to these country-level impacts, the global sum of targeted border closures implemented by February 5, 2020 was not sufficient to slow global COVID-19 transmission, but the sum of total border closures implemented by March 19, 2020 did achieve this effect. Country-level results were highly heterogeneous, with early implementation and border closures so broadly targeted that they resemble total border closures improving the likelihood of slowing the pandemic's spread. Governments that can make productive use of extra preparation time and cannot feasibly implement less restrictive alternatives might consider enacting border closures. However, given their moderate and uncertain impacts and their significant harms, border closures are unlikely to be the best policy response for most countries and should only be deployed in rare circumstances and with great caution. All countries would benefit from global mechanisms to coordinate national decisions on border closures during pandemics.

## Introduction

Before the COVID-19 pandemic, targeted border closures–restrictions on non-essential entry of foreign nationals from one or more specified countries–were not shown to be effective in

https://dataverse.scholarsportal.info/dataverse/CTR/.

**Funding:** SJH is supported by the Canadian Institutes of Health Research (#172982). The funder had no role in study design, data collection and analysis, decision to publish, or preparation of the manuscript.

**Competing interests:** I have read the journal's policy and the authors of this manuscript have the following competing interests: SJH is the Vice-President of Corporate Data & Surveillance at the Public Health Agency of Canada (PHAC) and previously served as the Scientific Director of the Institute of Population & Public Health at the Canadian Institutes of Health Research (CIHR). The views expressed in this article are those of the authors and do not necessarily reflect those of PHAC, CIHR or the Government of Canada.

controlling the spread of emerging infectious diseases such as influenza, Ebola, and other coronaviruses [1, 2]. Total border closures–restrictions on non-essential entry of foreign nationals from all countries–were rarely enacted prior to COVID-19 and, as such, little evidence has been available about their effectiveness [3]. Nevertheless, the unprecedented speed with which COVID-19 spread globally reignited a debate over whether targeted and/or total border closures can prevent or slow the international transmission of a pandemic. Despite quasi-experimental evaluations of the use of masks [4, 5] and physical distancing [5–7], there are very few quasi-experimental evaluations of whether national border closures reduced the global spread of COVID-19 in the initial months of the pandemic.

Prior evidence on the effectiveness of border closures to control the transmission of viral pathogens is mixed and largely derived from mathematical models [1, 2, 8]. One study found that targeted travel restrictions between the world's largest cities would result in a 50% reduction in the international transmission of SARS-CoV-1 [9], whereas a systematic review of influenza transmission concluded that restricting at least 90% of air travel would likely delay the spread of a pandemic by 3–4 weeks [1]. In the case of COVID-19, domestic travel restrictions appear to have slowed outbreaks in several contexts [7, 10–12], but there is mixed evidence of international impact [12]. Some mathematical models suggest border closures targeting China had little or no effect on global COVID-19 transmission due to the number of exportation events that had already occurred prior to their implementation [13–17], with one study identifying North America and Europe as the primary regions of origin for imported cases in other countries [18].

Although these mathematical models can provide valuable insights on the dynamics of disease transmission–particularly during the early stages of a public health event or when real-world data are scarce–we are now able to evaluate real-world data using quasi-experimental methods that do not rely on parameter assumptions which can vary among modelling studies [8, 19]. Two quasi-experimental studies of 235 country entities [20] and of nine African countries [21] have found that both targeted and total border closures have not had a significant impact on the international transmission of COVID-19. Others have found that early border closures can reduce exportation of cases and could have bought valuable weeks of preparation time if paired with further disease prevention measures [8, 12, 22–28].

We conducted quasi-experimental analyses to determine whether and under what conditions national border closures affected the transmission of COVID-19 within and across 166 countries during the first 22 weeks of the pandemic. This critical period in the emergence of the pandemic saw the rapid implementation of national border closures impacting at least 95% of the world's population, with reported global incidence eventually surpassing 30,000 cases per day (Fig 1). The remarkable speed and concomitance with which countries closed their borders means we may never have a better natural experiment to observe and measure the effects of this intervention. Rather than estimating the hypothetical effect of border closures in isolation of other control measures as in a mathematical modelling study, we conceptualize border closures as both a restriction on travel and a powerful social signal that potentially enhances the effect of complementary public health measures and posit that the real-world impact of these policies can never be fully disentangled from complementary measures (Fig A in S1 Text). Although data quality issues and concurrent implementation of various public health measures limit our ability to estimate a generalizable magnitude of effect, this impact evaluation study leverages an unprecedented natural experiment to produce a real-world assessment of whether and under what conditions national border closures may be effective in the early days of a pandemic. In doing so, this study also represents an important scientific advance in the use of quasi-experimental methods to better understand the impact of national public health interventions in a globalized world.

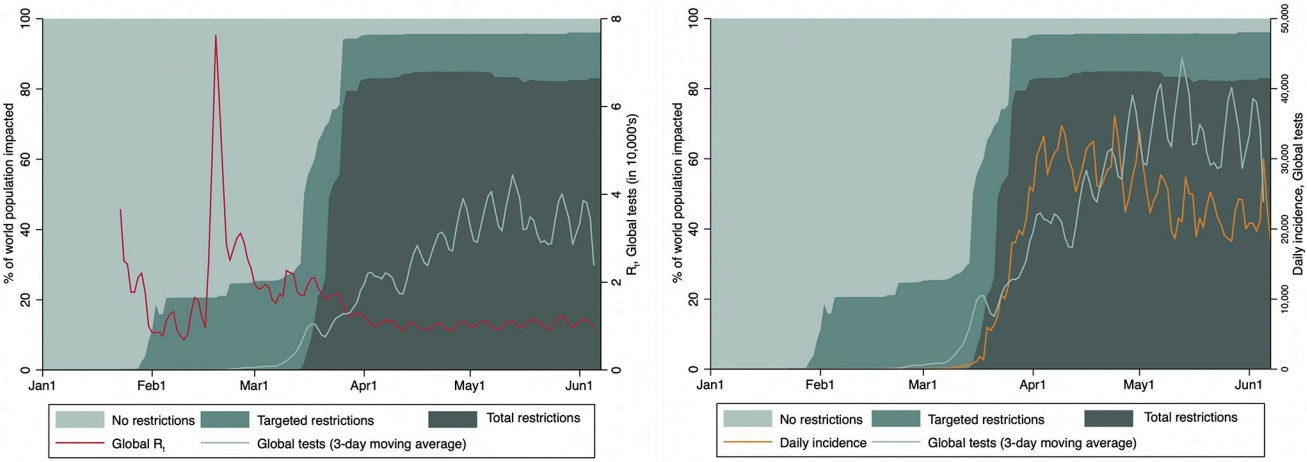

**Fig 1. Global border closures and changes in COVID-19 R_t and incidence.** Stacked area plot of the world's population living in countries with no border closures, targeted border closures, and total border closures. From left to right are the time-series trends of: i) global time-varying effective reproductive number ($R_t$) and number of global diagnostic tests; and ii) daily incidence and number of global diagnostic tests.

## Methods

Our multi-method quasi-experimental evaluation of the effect of border closures on the transmission of COVID-19 triangulates results across interrupted time-series (ITS) analysis, coarsened exact matching (CEM), and meta-regression (Fig B in S1 Text). Detailed descriptions of these analyses can be found in the Supplementary Information.

We selected the instantaneous effective reproduction number ($R_t$) as the primary outcome to produce a balanced weighting of both the exponential spread of initial outbreaks and large absolute increases in case counts in later weeks. As a measure of the expected number of new infections caused by one infected individual, an $R_t$ of less than one leads to a reduction in COVID-19 propagation, while an $R_t$ of greater than one leads to expansion of the pandemic [29]. Country-level, global, and stratified $R_t$ were independently calculated from daily incidence data, producing a stationary estimation of the transmission of COVID-19 during the first pandemic wave with limited serial correlation [30]. This means that longitudinal increases in testing could temporarily bias country-level $R_t$ upward, while global and stratified $R_t$ could be more sensitive to changes in incidence in countries with more robust testing regimes.

All countries for which Johns Hopkins University Center for Systems Science and Engineering (CSSE) COVID-19 incidence data and country-level covariates (Table A in S1 Text) could be obtained were included in analysis [31]. Border closure data for each country-pair (i.e. country implementing a border closure and each country targeted by the border closure) were obtained from the Oxford COVID-19 Government Response Tracker and recoded into matrix form for each date between the study period of January 1, 2020 to June 8, 2020 (S1 Data) [32]. The global intervention points of border closures were defined as the dates at which countries comprising at least 20% of the global population had implemented targeted border closures or total border closures, resulting in global intervention dates of February 5, 2020 for targeted closures and March 19, 2020 for total border closures. Detailed explanations on the selection of the primary outcome variable, quantitative methods, and scenario analyses are found in the Supplementary Information.

ITS analyses were conducted to evaluate the change in both level and slope of $R_t$ after border closures were implemented using varying levels of lagged outcomes, with 5-day lagged $R_t$ designated *a priori* as the primary measure [33]. We selected linear over exponential or logarithmic

regression models to more conservatively compare outcomes between country- and global-level time-series without inappropriately smoothing real-world trends [34]. By fitting least squares regression lines to pre- and post-border closure data, the analyses identify statistically significant changes in COVID-19 transmission at both the country and global level assuming a constant underlying time trend in $R_t$. Meta-regression of ITS results were then conducted using ordered logistic regression to identify which *a priori* country-level factors hypothesized to be associated with effective national border closures based on published literature (Table A in S1 Text) were associated with positive border closure effects.

In a separate analysis to triangulate results, CEM was used to reduce the multivariate imbalance between countries without border closures (control group) and countries with a border closure (treatment group). Countries were assigned treatment status using two different approaches and three models of theoretically sampled coarsened covariates; an adjusted regression analysis using maximum likelihood estimation was then used to quantify the effect of border closures on $R_t$ [35] This matching process improves estimates by reducing selection bias, resulting in an empirical distribution that is more representative to draw inferences of the effect of border closures on $R_t$ [35, 36].

An extensive series of 328 robustness checks are detailed in the text of the Supplementary Information, including falsification tests using placebo intervention dates (Table B in S1 Text), aligning analysis by time of intervention (Tables C, D in S1 Text), censoring extreme values (Tables E, F in S1 Text), restricting pre- and post-intervention data points (Tables G, H in S1 Text), and stratifying analysis for travel quarantine-implementing countries (Tables I, J in S1 Text).

## Results

### Global-level border closure analyses

ITS analysis identifies a stable reduction in the additional number of global cases expected for each case of COVID-19 following the implementation of total border closures, whereas the use of targeted border closures did not lead to a reduction in the global transmission of COVID-19 (Fig 2). Both the level (2.36; CI: 1.18–3.53) and slope (0.10; CI: 0.04–0.16) of $R_t$ significantly increased as compared to the underlying time trend after the first wave of targeted border closures on February 5, 2020 (Table 1). In contrast, there was a significant drop in the level (-0.62; CI: -1.05 –-0.19) of the global $R_t$ following the wave of total border closures on March 19, 2020, as compared to the first wave of targeted closures. An alternate model aligning ITS analysis by intervention date for all countries that only implemented total border closures also indicates a statistically significant and visually apparent reversal in slope (-0.09; CI: -0.10 –-0.07) of the $R_t$ (Fig 1; Tables C, D in S1 Text). Disaggregating analysis by high-income country (HIC) versus low- and middle-income country (LMIC) status did not reveal any major differences (Fig C in S1 Text; Tables K-P Tables in S1 Text).

Global results were robust to lagged effects up to 14 days for the first intervention and up to six days for the second intervention (Table Q in S1 Text) and were not affected by including China in the analysis (Tables R, S in S1 Text), by censoring extreme Rt values from February 18–21, 2020 (Tables E, F in S1 Text), or by restricting analysis to 45 days pre- and post-intervention for both targeted and total closures (Tables G, H in S1 Text). Further sensitivity checks addressing the potential confounding effect of the concurrent implementation of co-interventions (Fig 1; Tables I, J in S1 Text) resulted in nearly identical decreases in Rt following total border closures among countries that implemented travel quarantine measures (-0.82; SE = 0.21) and countries that had not implemented quarantine measures (-0.84; SE = 0.24).

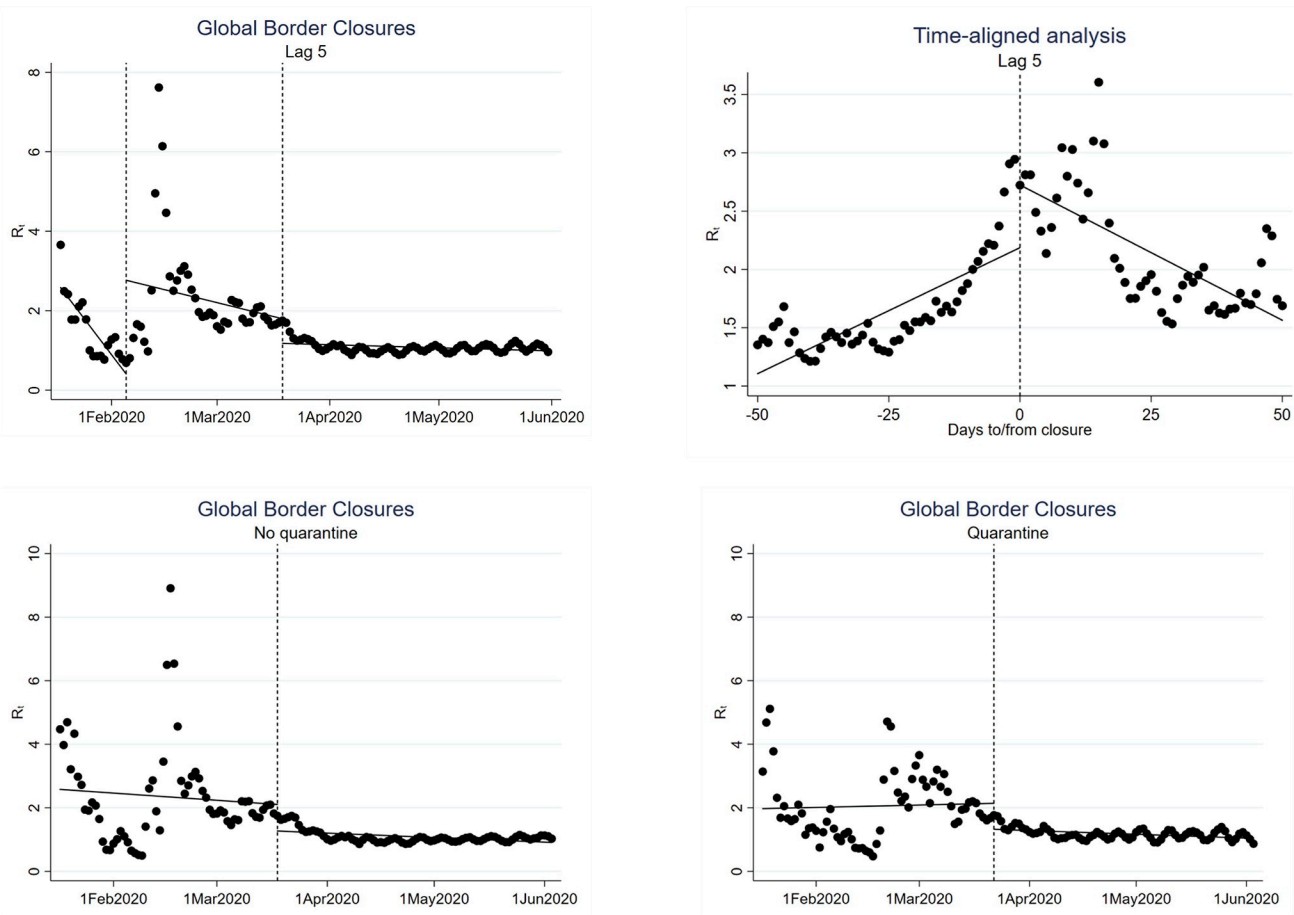

**Fig 2. Interrupted time-series analyses of global $R_t$, excluding China.** Time-series analyses of: i) impact of the first global intervention of targeted border closures on February 5, 2020, followed by the second global intervention of total border closures on March 19, 2020; ii) time-aligned analysis of the intervention date for all countries that only implemented total border closures; iii) impact of the global intervention of total border closures for countries that had not implemented quarantine measures for international travellers (March 18, 2020); and iv) impact of the global intervention of total border closures for countries that had implemented quarantine measures for international travellers (March 22, 2020).

## Country-level border closure analyses

Country-level ITS results were highly heterogeneous, with beneficial (i.e., decreased $R_t$) and null effects generally more common than adverse (i.e., increased Rt) and mixed effects (Fig 3). Although total border closures were more likely to reduce COVID-19 transmission rates than targeted border closures, both interventions were marked by significant heterogeneity in their outcomes (Table T in S1 Text). Of the 34 targeted border closures that could be analyzed, countries were about as likely to experience beneficial effects (decreased $R_t$ in 12 countries), adverse effects (increased Rt in 10 countries), or null or mixed effects (12 countries), while the 103 total border closures that could be analyzed resulted in more beneficial effects (41 countries) than adverse effects (28 countries), while 34 countries experienced null or mixed effects. Of the five analyzed countries that partially reopened from a total border closure to targeted border closures, two experienced increased COVID-19 transmission rates and three had null effects.

Similar to global results, country-level effects were robust to varying lags in outcome measures (S2 Data). Falsification tests for both global- and country-level ITS analyses using

**Table 1. Global interrupted time-series results.** Analyses with two intervention points on February 5, 2020 and March 19, 2020, with lags from 0–15 days. Coefficients, lower 95% confidence interval (LCI), upper 95% confidence interval (UCI), and standard errors (SE) are provided for the underlying time trend, two slope and level changes, and a constant. Analyses exclude China.

|  | 0-day lag | 1-day lag | 2-day lag | 3-day lag | 4-day lag | 5-day lag | 6-day lag | 7-day lag |
|---|---|---|---|---|---|---|---|---|
| Time | -0.200*** | -0.177*** | -0.154*** | -0.134*** | -0.126*** | -0.121*** | -0.116*** | -0.110*** |
| LCI | -0.258 | -0.24 | -0.218 | -0.196 | -0.179 | -0.167 | -0.157 | -0.148 |
| UCI | -0.141 | -0.114 | -0.09 | -0.0725 | -0.073 | -0.0743 | -0.0751 | -0.0715 |
| SE | 0.0296 | 0.0319 | 0.0324 | 0.0312 | 0.0269 | 0.0234 | 0.0208 | 0.0193 |
| Feb 5 level change | 1.776*** | 1.866*** | 1.911*** | 1.951*** | 2.139*** | 2.359*** | 2.593*** | 2.786*** |
| LCI | 0.83 | 0.823 | 0.787 | 0.77 | 0.957 | 1.183 | 1.436 | 1.648 |
| UCI | 2.721 | 2.909 | 3.035 | 3.133 | 3.321 | 3.534 | 3.751 | 3.923 |
| SE | 0.478 | 0.527 | 0.568 | 0.597 | 0.598 | 0.594 | 0.585 | 0.575 |
| Feb 5 slope change | 0.205*** | 0.177*** | 0.150*** | 0.125*** | 0.110*** | 0.0981*** | 0.0866*** | 0.0731*** |
| LCI | 0.141 | 0.109 | 0.0792 | 0.0554 | 0.0476 | 0.0403 | 0.0327 | 0.0214 |
| UCI | 0.268 | 0.246 | 0.22 | 0.194 | 0.173 | 0.156 | 0.14 | 0.125 |
| SE | 0.0321 | 0.0347 | 0.0357 | 0.0351 | 0.0317 | 0.0292 | 0.0272 | 0.0261 |
| Mar 19 level change | -1.014*** | -0.958*** | -0.894*** | -0.815*** | -0.718*** | -0.620*** | -0.523** | -0.422* |
| LCI | -1.38 | -1.32 | -1.268 | -1.214 | -1.135 | -1.051 | -0.961 | -0.866 |
| UCI | -0.648 | -0.596 | -0.52 | -0.415 | -0.3 | -0.189 | -0.0851 | 0.0212 |
| SE | 0.185 | 0.183 | 0.189 | 0.202 | 0.211 | 0.218 | 0.221 | 0.224 |
| Mar 19 slope change | -0.01 | -0.00472 | 0.00048 | 0.00587 | 0.0127 | 0.02 | 0.0277 | 0.0354** |
| LCI | -0.035 | -0.0318 | -0.0289 | -0.0259 | -0.0208 | -0.0145 | -0.00716 | 0.00051 |
| UCI | 0.0149 | 0.0223 | 0.0298 | 0.0376 | 0.0461 | 0.0545 | 0.0626 | 0.0702 |
| SE | 0.0126 | 0.0137 | 0.0148 | 0.016 | 0.0169 | 0.0175 | 0.0176 | 0.0176 |
| Constant | 2.931*** | 2.841*** | 2.742*** | 2.649*** | 2.608*** | 2.579*** | 2.554*** | 2.515*** |
| LCI | 2.441 | 2.331 | 2.215 | 2.112 | 2.09 | 2.076 | 2.065 | 2.029 |
| UCI | 3.422 | 3.35 | 3.269 | 3.187 | 3.127 | 3.082 | 3.043 | 3.001 |
| SE | 0.248 | 0.257 | 0.266 | 0.272 | 0.262 | 0.254 | 0.247 | 0.246 |
| Observations | 135 | 135 | 135 | 135 | 135 | 135 | 135 | 135 |
|  | **8-day lag** | **9-day lag** | **10-day lag** | **11-day lag** | **12-day lag** | **13-day lag** | **14-day lag** | **15-day lag** |
| Time | -0.097*** | -0.082*** | -0.070*** | -0.065*** | -0.062*** | -0.046** | -0.013 | 0.032 |
| LCI | -0.138 | -0.126 | -0.114 | -0.104 | -0.0974 | -0.0886 | -0.0821 | -0.0673 |
| UCI | -0.0559 | -0.0375 | -0.0267 | -0.0248 | -0.0258 | -0.0028 | 0.0555 | 0.132 |
| SE | 0.0207 | 0.0224 | 0.022 | 0.02 | 0.0181 | 0.0217 | 0.0348 | 0.0504 |
| Feb 5 level change | 2.840*** | 2.817*** | 2.828*** | 2.951*** | 3.135*** | 2.939*** | 2.180*** | 0.855 |
| LCI | 1.675 | 1.608 | 1.621 | 1.811 | 2.119 | 1.845 | 0.683 | -1.086 |
| UCI | 4.005 | 4.025 | 4.036 | 4.09 | 4.15 | 4.033 | 3.677 | 2.796 |
| SE | 0.589 | 0.611 | 0.611 | 0.576 | 0.513 | 0.553 | 0.757 | 0.981 |
| Feb 5 slope change | 0.0540* | 0.0336 | 0.0161 | 0.00324 | -0.00735 | -0.026 | -0.053 | -0.0842 |
| LCI | -6.80E-05 | -0.0237 | -0.0407 | -0.0496 | -0.0547 | -0.0789 | -0.129 | -0.186 |
| UCI | 0.108 | 0.0908 | 0.0728 | 0.056 | 0.04 | 0.0269 | 0.023 | 0.0178 |
| SE | 0.0273 | 0.0289 | 0.0287 | 0.0267 | 0.0239 | 0.0267 | 0.0384 | 0.0516 |
| Mar 19 level change | -0.312 | -0.194 | -0.0671 | 0.0736 | 0.22 | 0.294 | 0.25 | 0.0749 |
| LCI | -0.776 | -0.684 | -0.574 | -0.427 | -0.241 | -0.176 | -0.24 | -0.255 |
| UCI | 0.151 | 0.295 | 0.44 | 0.574 | 0.681 | 0.765 | 0.739 | 0.405 |
| SE | 0.234 | 0.247 | 0.256 | 0.253 | 0.233 | 0.238 | 0.248 | 0.167 |
| Mar 19 slope change | 0.0420** | 0.0478** | 0.0539*** | 0.0612*** | 0.0693*** | 0.0724*** | 0.0673*** | 0.0530*** |
| LCI | 0.00656 | 0.0116 | 0.0175 | 0.0263 | 0.0382 | 0.0414 | 0.0349 | 0.031 |
| UCI | 0.0774 | 0.084 | 0.0903 | 0.0961 | 0.1 | 0.103 | 0.0996 | 0.0749 |
| SE | 0.0179 | 0.0183 | 0.0184 | 0.0176 | 0.0157 | 0.0157 | 0.0164 | 0.0111 |

*(Continued)*

**Table 1.** (Continued)

| | | | | | | | | |
|---|---|---|---|---|---|---|---|---|
| **Constant** | 2.433*** | 2.334*** | 2.253*** | 2.210*** | 2.188*** | 2.061*** | 1.791*** | 1.395** |
| **LCI** | 1.921 | 1.788 | 1.693 | 1.66 | 1.652 | 1.463 | 0.986 | 0.298 |
| **UCI** | 2.945 | 2.88 | 2.813 | 2.761 | 2.724 | 2.658 | 2.595 | 2.491 |
| **SE** | 0.259 | 0.276 | 0.283 | 0.278 | 0.271 | 0.302 | 0.407 | 0.554 |
| **Observations** | 135 | 135 | 135 | 135 | 135 | 135 | 135 | 135 |

*** $p < 0.01$,

** $p < 0.05$,

* $p < 0.10$

## Targeted border closures

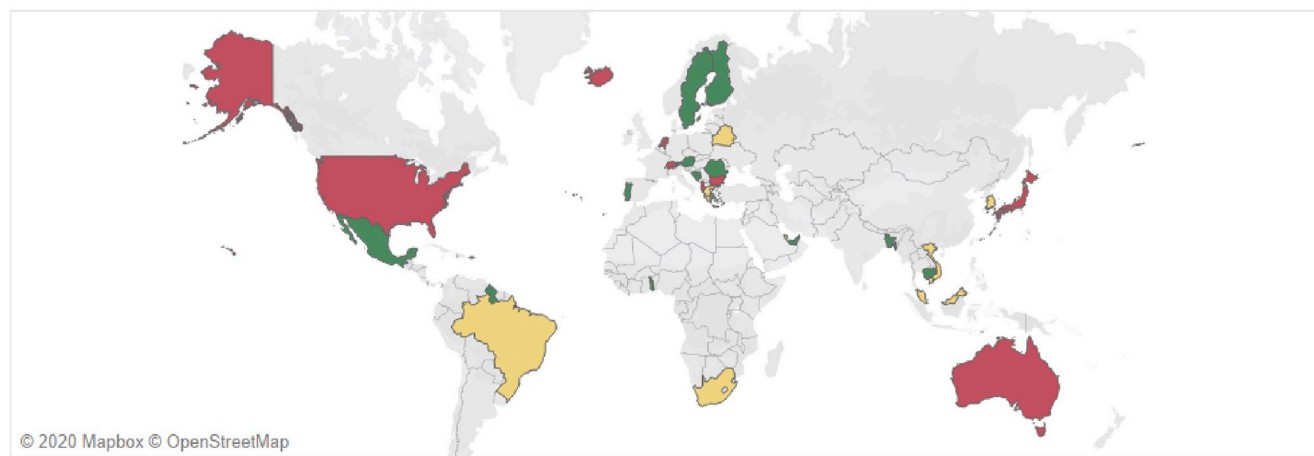

## Total border closures

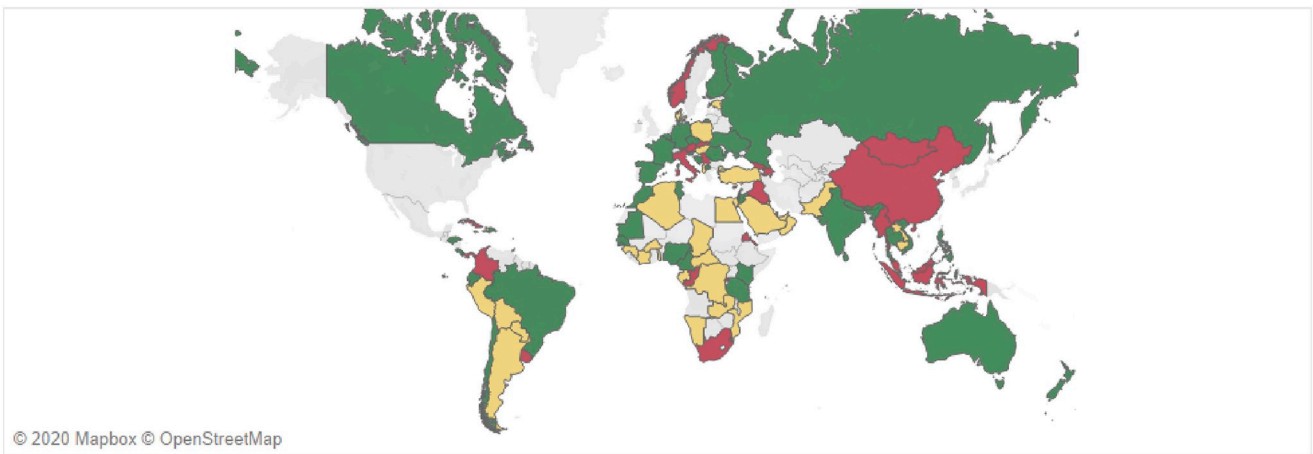

**Fig 3. Country-specific ITS results for targeted and total border closures.** Map of targeted (top) and total (bottom) border closures that produced positive effects (i.e., decreased Rt) (green), negative effects (i.e., increased $R_t$) (red), and null or mixed effects (yellow) for all five-day lagged evaluations using ITS analysis. Map base layer from © OpenStreetMap can be accessed at https://www.openstreetmap.org.

placebo intervention dates for global and country-specific ITS analyses provide higher confidence that the observed effects were not due to chance, with the statistically significant change in the level of global transmission becoming statistically insignificant, the direction of country-specific time-series reversing from positive to negative for a majority of countries, and the magnitude of effect sizes decreasing by 56% (Table U in S1 Text).

Meta-regression analysis of country-level ITS results using a priori selected factors points to late implementation of border closures (after March 21) as the most consistently statistically significant adverse factor for both total and targeted border closures (Tables V, W in S1 Text). The majority of the 31 country-level factors tested, including economic indicators, gender parity, health metrics, domestic containment indicators, and border closure characteristics, had no statistically significant association with the likelihood of a national border closure having a beneficial effect. Factors associated with more effective total border closures were higher Global Health Security Index scores (-0.03, SE = 0.01), larger population (-0.29 (logged), SE = 0.12), and not being among the last third of countries to implement a border closure (1.48, SE = 0.46). Targeted border closures were less likely to have beneficial effects in countries with higher health expenditures (29.09, SE = 12.26).

## Coarsened exact matching analyses

CEM results suggest that a synthetic country with socio-economic and political characteristics similar to the Philippines or Colombia implementing a total border closure could expect their $R_t$ to decrease by 1.2 under conditions similar to COVID-19's first pandemic wave (Table 2). Although this would imply a reduction in COVID-19 transmission of 51% for a country that had an average $R_t$ of 2.4 on March 19, 2020, this estimated reduction in transmission is likely biased upward by an increase in testing capacity prior to this date [37, 38]. If this synthetic country were to implement a targeted border closure, it would likely experience a decrease in $R_t$ of 0.8, with a greater probability of decreasing transmission by increasing the number of countries targeted, regardless of whether those countries were known to have high levels of COVID-19 incidence at that time.

Multiple robustness checks were conducted, including varying the degree to which covariates were coarsened (Tables X, Y in S1 Text), using different combinations of coarsened covariates (Table Z in S1 Text), and restricting data points to 45 days or 60 days pre- and post-intervention (Tables AA, AB in S1 Text). Furthermore, scenario analyses splitting countries into high or low covariate distributions in social, economic, and political country-level factors prior to matching did not alter the statistically significant beneficial effects of both targeted and total border closures across all sets of analyses (Tables AC, AD in S1 Text). Finally, a sensitivity check matching countries on potentially confounding co-interventions, including workplace and public transit closures, stay at home orders, restrictions on public gatherings, and restrictions on internal movement did not significantly alter results (Tables AE, AG in S1 Text).

## Discussion

Two consistent findings emerge from this quasi-experimental evaluation of the effect of national border closures on COVID-19 transmission. First, total border closures and (to a lesser extent) targeted border closures had some effect on temporarily slowing COVID-19 transmission. Second, while the global sum of *targeted* border closures implemented by February 5, 2020 were not sufficient to slow the COVID-19 pandemic, the global sum of *total* border closures implemented by March 19, 2020 did result in a statistically significant reduction in global COVID-19 transmission.

**Table 2. Coarsened exact matching results.** Maximum likelihood random-effects estimation for targeted and total border closures. Country-days are matched on coarsened GHSI score, logged GDP, gender parity score, emigrants per capita, democracy index, logged airline passengers, health expenditures, and proportion of female government ministers.

| | Targeted border closures | | | |
| --- | --- | --- | --- | --- |
| | Dropping countries with total border closures | Controlling for countries with total border closures | Model using % of global population targeted | Model using % of global cases targeted |
| Targeted closure (β) | -0.73*** (0.06) | -0.67*** (0.04) | | |
| Confidence interval | [-0.84, -0.61] | [-0.75, -0.59] | | |
| Total closure (β) | | -1.26*** (0.03) | | |
| Confidence interval | | [-1.33, -1.20] | | |
| % of global pop targeted (β) | | | -1.30*** (0.09) | |
| Confidence interval | | | [-1.48, -1.12] | |
| % of global cases targeted (β) | | | | -0.20*** (0.07) |
| Confidence interval | | | | [-0.35, -0.06] |
| Constant | 2.38*** (0.05) | 2.50*** (0.04) | 2.27*** (0.04) | 2.16*** (0.05) |
| Confidence interval | [2.28, 2.48] | [2.42, 2.59] | [2.19, 2.36] | [2.06, 2.26] |
| Observations | 4,550 | 10,731 | 4,550 | 4,550 |
| Countries | 109 | 112 | 109 | 109 |
| | Total border closures | | | |
| | Primary Model | | Conservative Model | |
| | Dropping countries with targeted border closures | Controlling for countries with targeted border closures | Dropping countries with targeted border closures | Controlling for countries with targeted border closures |
| Targeted closure (β) | | -0.85*** (0.06) | | -0.66*** (0.05) |
| Confidence interval | | [-0.96, -0.74] | | [-0.77, -0.56] |
| Total closure (β) | -1.24*** (0.05) | -1.35*** (0.05) | -0.91*** (0.04) | -0.94*** (0.04) |
| Confidence interval | [-1.34, -1.15] | [-1.45, -1.25] | [-0.99, -0.83] | [-1.03, -0.86] |
| Constant | 2.34*** (0.05) | 2.37*** (0.06) | 2.31*** (0.04) | 2.372*** (0.04) |
| Confidence interval | [2.24, 2.44] | [2.26, 2.49] | [2.23, 2.38] | [2.29, 2.46] |
| Observations | 3,409 | 4,835 | 5,321 | 7,488 |
| Countries | 45 | 46 | 98 | 112 |

*** $p < 0.01$,

** $p < 0.05$,

* $p < 0.10$;

standard errors in parentheses

Although this study provides some evidence to support the use of national border closures to temporarily slow the first wave of a pandemic, much of their value will depend on how governments make use of the extra preparation time that these emergency interventions buy as weighed against their significant economic and social consequences [1, 2, 15]. More specifically, the potential for a moderate temporary reduction in domestic $R_t$ must be weighed by governments against the considerable negative downstream consequences of border closures and the possibility that they may not work or may even increase transmission in some implementing countries. For example, there is evidence to suggest that increased repatriation of emigrants immediately following border closures may have temporarily increased transmission in some countries [39, 40]. These dynamics may have contributed to the absence of difference in border closure effectiveness between HIC and LMIC observed in our study, but the

relatively even 60–75% reduction in airline travel across regions during the study period indicates that border closures were meaningfully enforced across all country income levels [41].

Less restrictive alternatives, such as mandatory quarantine for all incoming travellers, might prove to be equally effective and less harmful for those countries with the bureaucratic capacity and political will to fully implement and enforce them [42]. Nevertheless, border closures may be helpful in those rare cases when countries are unable to implement mandatory quarantines, are willing to adopt these measures early, and where the extra time bought by moderate temporary reductions in viral transmission is determined to be worth border closures' significant trade-offs and consequences. With national border closures contributing to increases in unemployment [43], higher costs of goods and services [44], and a historic reduction in economic growth [45, 46] across the world, these consequences should not be taken lightly.

Globally, the cumulative impacts of total border closures appeared to reduce COVID-19 transmission more effectively than the cumulative impacts of targeted border closures. Indeed, this study's CEM analysis provides evidence of a dose-response relationship in the percentage of the world's population that targeted border closures affect, wherein the most effective targeted border closures were those that functionally approximated total border closures by limiting travel from nearly all countries. Although our findings run counter to null findings from a panel-matching analysis of border closures' impacts on the global spread of COVID-19 [20], they do align with significant impacts of early and more comprehensive border closures from the largest systematic review of the subject [8]. These divergent findings can be attributed to a number of differences in model specification, data aggregation, covariate selection, and most notably, the use of smoothed per capita COVID-19 incidence as the primary outcome [20].

While border closures in the initial weeks of COVID-19 likely violated Article 43 of the legally-binding International Health Regulations because they were implemented without "available scientific evidence" [42, 47, 48], our study's evidence of their potential effectiveness in very specific circumstances points toward the need to reconsider past blanket guidance against border closures and take a more nuanced approach to their potential deployment. Most specifically, with increasing evidence that border closures played a moderate role in slowing the initial spread of COVID-19 and buying some additional time to prepare for the oncoming pandemic [8], public health authorities should re-evaluate their guidance so that it is more context-specific and consider global mechanisms to coordinate national decisions on border closures. This increasing evidence about the moderate effectiveness of border closures should also be reflected in future revisions to the International Health Regulations and in negotiations towards a pandemic instrument [49–51].

The numerous co-interventions, behavioural adaptations, and improvements in data quality occurring throughout the January 1-June 8, 2020 study period call for caution in the direct interpretation of point estimates of effect sizes. We attempted to minimize biases and reduce confounding by: evaluating at multiple levels (i.e., global, income group, and country); triangulating results from time-series and synthetic control quasi-experimental designs; and conducting an extensive series of falsification, robustness and sensitivity checks using alternative models, time frames and lags. Yet, the possibility of time-dependent misreporting, limited data availability, and persistent under-detection of COVID-19 incidence remain. Because of these unavoidable challenges with analyzing data from initial pandemic waves, in this article we emphasize the consistent directional findings of total border closures' greater effects on COVID-19 transmission than targeted border closures, rather than potentially inaccurate point estimates of border closures' impacts on $R_t$.

Nevertheless, if underreporting of cases within countries can be assumed to have been more prevalent in the initial stages of the pandemic, our findings of decreased propagation at

the later stages of the study period can be considered conservative. In addition, the second global intervention evaluated (i.e., total border closures) was found to have a statistically significant protective effect in spite of late-adopting countries being less likely to effectively implement closures. Finally, it is important to note that this study focused on the implementation of border closures during the first phase of the COVID-19 pandemic, did not investigate the impact of maintaining, lifting or reimposing border closures when many countries faced subsequent pandemic waves, and may not be generalizable to other infectious diseases. Indeed, in the year following our study period, targeted border closures enacted in response to the emergence of the Omicron variant of COVID-19 failed to prevent its rapid global dissemination [52, 53].

## Conclusions

This study leverages an unprecedented natural experiment to conduct quasi-experimental analyses of border closures' impacts on global viral transmission. Analyses consistently point to stronger effects of total border closures over targeted border closures to reduce transmission levels of COVID-19 in the initial phases of the pandemic. Although this study identifies earlier adoption and a greater proportion of global population targeted as being important determinants of border closures' effectiveness, it is difficult to offer concrete guidance for countries that hope to implement targeted border closures through proposed COVID-19 "travel corridors" or restricting travel to countries that have identified new SARS-CoV-2 variants of concern. Such guidance would need to be tailored to the specific country context, the threat being addressed, the feasibility of implementation, and the availability of less restrictive alternatives. The best available evidence on infectious disease border closures has changed as a result of the COVID-19 pandemic. Governments and international agencies should consider the implications of these findings for both their pandemic responses and in designing new global governance arrangements that structure them [54].

In the meantime, governments that were willing to endure the negative economic and social consequences of total or near-total border closures may have succeeded in temporarily slowing COVID-19 transmission in the first phase of the pandemic, which could have bought some countries additional time to implement more robust public health measures. However, unless governments made productive use of this extra preparation time, slowing the COVID-19 pandemic's arrival might not have been worth the immense consequences of border closures. Also, future border closures are unlikely to impact the global transmission of a pandemic unless the unprecedented scale of these simultaneous border closures impacting over 95% of the world's population can be repeated. Overall, border closures are not panaceas and should only be deployed in rare circumstances and with great caution.

## Supporting information

**S1 Text.**
(PDF)

**S1 Data. Matrix of border-closure status by country-day.** Travel restriction status by country-day of every border closure included in the study analyses [available as csv on Scholars Portal].
(XLSX)

**S2 Data. Country-specific interrupted time-series results.** Results presented for 0 to 15-day lags [available as csv on Scholars Portal].
(XLSX)

**S3 Data. Analysis plan.** Ex ante analysis plan with changes to analytical approach noted [available as pdf on Scholars Portal].
(DOCX)

**S4 Data. Global and country-specific interrupted time-series results.** Dickey-Fuller tests for unit roots, Cumby-Huizinga tests for autocorrelation, and results correcting for each lag found to have serial correlation present for all global and country interrupted time series [available as csv on Scholars Portal].
(XLSX)

## Acknowledgments

We gratefully acknowledge Gary King for comments on methodological aspects of the manuscript, Michèle Palkovits and Isaac Weldon for their contributions to interpretation of results, as well as Tina Nanyangwe-Moyo and Zun Ge Mao for their contributions to literature review and manuscript revision.

## Author Contributions

**Conceptualization:** Mathieu J. P. Poirier, Susan Rogers Van Katwyk, Steven J. Hoffman.

**Data curation:** Gigi Lin.

**Formal analysis:** Mathieu J. P. Poirier, Gigi Lin.

**Funding acquisition:** Steven J. Hoffman.

**Investigation:** Mathieu J. P. Poirier, Susan Rogers Van Katwyk, Gigi Lin, Steven J. Hoffman.

**Methodology:** Mathieu J. P. Poirier, Susan Rogers Van Katwyk, Gigi Lin, Steven J. Hoffman.

**Project administration:** Mathieu J. P. Poirier.

**Supervision:** Mathieu J. P. Poirier, Steven J. Hoffman.

**Validation:** Mathieu J. P. Poirier, Susan Rogers Van Katwyk, Gigi Lin, Steven J. Hoffman.

**Visualization:** Mathieu J. P. Poirier, Gigi Lin.

**Writing – original draft:** Mathieu J. P. Poirier, Susan Rogers Van Katwyk, Gigi Lin, Steven J. Hoffman.

**Writing – review & editing:** Mathieu J. P. Poirier, Susan Rogers Van Katwyk, Gigi Lin, Steven J. Hoffman.

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
