## [Decision Letter · Decision Letter 0]

24 Jun 2022

PGPH-D-21-01124

Quasi-experimental evaluation of national border closures on COVID-19 transmission

Dear Dr. Hoffman,

Thank you for submitting your manuscript to PLOS Global Public Health. After careful consideration, we feel that it has merit but does not fully meet PLOS Global Public Health’s publication criteria as it currently stands. Therefore, we invite you to submit a revised version of the manuscript that addresses the points raised during the review process.

We look forward to receiving your revised manuscript.

Kind regards,

Atanu Bhattacharjee, Ph.D

Academic Editor

Journal Requirements:

1. Please send a completed 'Competing Interests' statement, including any COIs declared by your co-authors. If you have no competing interests to declare, please state "The authors have declared that no competing interests exist". Otherwise please declare all competing interests beginning with the statement "I have read the journal's policy and the authors of this manuscript have the following competing interests:"

2. Please provide separate figure files in .tif or .eps format and removed from the manuscript file.

3. Figure 3: please (a) provide a direct link to the base layer of the map used and ensure this is also included in the figure legend; (b) provide a link to the terms of use / license information for the base layer. We cannot publish proprietary or copyrighted maps (e.g. Google Maps, Mapquest) and the terms of use for your map base layer must be compatible with our CC-BY 4.0 license. 

Additional Editor Comments (if provided):

Authors are requested to modify the manuscript as suggested by the reviewers.

Reviewers' comments:

Reviewer's Responses to Questions

**Comments to the Author**

1. Does this manuscript meet PLOS Global Public Health’s publication criteria? Is the manuscript technically sound, and do the data support the conclusions? The manuscript must describe methodologically and ethically rigorous research with conclusions that are appropriately drawn based on the data presented.

Reviewer #1: Yes

Reviewer #2: Yes

2. Has the statistical analysis been performed appropriately and rigorously?

Reviewer #1: Yes

Reviewer #2: Yes

3. Have the authors made all data underlying the findings in their manuscript fully available (please refer to the Data Availability Statement at the start of the manuscript PDF file)?

Reviewer #1: Yes

Reviewer #2: Yes

4. Is the manuscript presented in an intelligible fashion and written in standard English?

Reviewer #1: Yes

Reviewer #2: Yes

5. Review Comments to the Author

Reviewer #1: Comments:

This is an interesting piece using the quasi-experimental analysis to evaluate the impact on transmission of COVID-19 by closure of national borders. While not mentioned, it could also be noted that the piece is timely because with new emerging variants of COVID-19, global cases are still on the rise and proper measures and models for public health interventions are required. The authors have also factored in the biases for covid-19 transmission and border closure.

However, there are a few minor areas in which I feel the paper needs some revision:

1. Line 30, Consider removing ‘but no’ instead use “Still, no”.

2. Line 38. Replace ‘not sufficient’ with insufficient.

3. Line 47. Consider deleting ‘decisions on’.

4. The authors have talked about effective reproductive number. As per Cori et al 2013 and Fraser C 2007, the effective reproduction number can be defined in 2 ways: instantaneous reproductive number or as the case reproductive number. Kindly elaborate on which of the two you have considered. I believe that you are considering instantaneous reproductive number but mention this in the text as well.

5. Line 144: Figure 2 is cited before figure 1 (line 151). Kindly correct this.

6. Can you elaborate on the impact of international border closures for only the top 5 hit nations like USA, India, UK, France etc. in 2021 (the second wave) in a short paragraph in discussion?

7. What could be the reason that no major disparities were observed between high-income vs low- and middle-income countries? please shed some light on it.

8. Line 184-190- Out of the countries that had adverse effects of border closures, how frequent the travel is from and to these countries? also, were these countries showed similar outcomes in the following ravel bans in other covid-19 waves as well.

9. The biases involved with border closures and Rt has been mentioned properly in the text.

10. Lines 303-306. Consider rephrasing to “This study presents the first quasi-experimental analysis of border closures’ impacts on viral transmission during a pandemic, taking advantage of an unprecedented natural experiment.”

11. Lines 307-311- add ‘a’ before greater proportion.

12. Lines 311-313- add ‘the’ before ‘threat’, ‘feasibility’ and availability.

13. Line 317. Consider removing ‘that were’ after government.

Reviewer #2: The research work described in this paper is commendable considering the time, energy and quantum of work input in this work. The manuscript is technically sound, systematic and meet PGPH publication criteria. The analysis is sound and all needed supporting information have been made available. However, there are a few comments and questions to be attended to by the authors.

Introduction: Please kindly provide a global picture of COVID-19 within the first 22 weeks of the pandemic while supporting your literature with figures.

Methodology: Line 75. Please what criteria was used in the selection of the 166 countries? Please state the inclusion criteria in the main work. Any justification for limiting your quasi-experimental study to the first 22 weeks of the COVID-19 Pandemic?

What sampling method and technique informed your selection of the 166 countries? Please provide a justification for your choice of the sampling method.

Line 243-244: Please operationally define 'total' and 'targeted' at their first appearance in the text.

Line 240-246: Findings well discussed but please kindly relate your findings to the findings of other studies and determine whether they share similarities or dissimilarities.

Line 263- 264: Please kindly cite an intext reference for that statement since that doesn't seem to reflect a finding from your study.

Line 288: Please kindly check the font size used

Line 294: Please check and correct the font and size.

Results: Please kindly provide a caption or label for figure 1

6. PLOS authors have the option to publish the peer review history of their article (what does this mean?). If published, this will include your full peer review and any attached files.

**Do you want your identity to be public for this peer review?** For information about this choice, including consent withdrawal, please see our Privacy Policy.

Reviewer #1: **Yes: **Rini Chaturvedi

Reviewer #2: **Yes: **Dorothy Serwaa Boakye

---

## [Decision Letter · Decision Letter 1]

17 Jan 2023

PGPH-D-21-01124R1

Quasi-experimental evaluation of national border closures on COVID-19 transmission

Dear Dr. Hoffman,

Thank you for submitting your manuscript to PLOS Global Public Health. After careful consideration, we feel that it has merit but does not fully meet PLOS Global Public Health’s publication criteria as it currently stands. Therefore, we invite you to submit a revised version of the manuscript that addresses the points raised during the review process.

EDITOR: Please insert comments here and delete this placeholder text when finished. Be sure to:

Indicate which changes you require for acceptance versus which changes you recommendAddress any conflicts between the reviews so that it's clear which advice the authors should followProvide specific feedback from your evaluation of the manuscript

Please ensure that your decision is justified on PLOS Global Public Health’s publication criteria and not, for example, on novelty or perceived impact.

We look forward to receiving your revised manuscript.

Kind regards,

Lorena G Barberia

Section Editor

Journal Requirements:

Additional Editor Comments (if provided):

Dear authors,

Thank you for submitting your research to PLOS Global Public Health. We requested an additional review of your revised manuscript that evaluates the impact of border-closure of 166 countries’ using multiple quasi-experimental methods. 

We would like for you to revise the final version of your manuscript to address the following minor points in order to strengthen the manuscript prior to publication. 

1. The role of mathematical models versus observational data. 

As the reviewer notes, some of the statements made in the Abstract and Introduction section imply that results from mathematical models are somewhat limited in evaluating the impact of border closures. Please remove the reference to mathematical models in the abstract. It is not central to the contribution of the manuscript. In the introduction, it would be helpful if you could address the contributions of using real-world data, but also acknowledge that mathematical models can be valuable during the early stages of a public health event (especially when real-world data are scarce). 

2. The dependent variable, Rt.

It would be helpful to note that different countries adopted different methodologies to count cases and testing supplies were limited early in the pandemic. Both of these issues affects the calculation of Rt. You note this in the discussion, but I think it could be acknowledged earlier on to address the validity of this variable as your preferred dependent variable. 

3. Total border closures banning vs. targeted border closures banning travel from specific countries 

A central contribution of the study is its capacity to show differences/tradeoffs from total border closures to targeted border closures, but as the study acknowledges total border closures only became more widespread after March 2020. The abstract states "Total border closures banning non-essential travel from all countries and (to a lesser extent) targeted border closures banning travel from specific countries had some effect on temporarily slowing COVID-19 transmission in those countries that implemented them." I think the abstract could emphasize the differences in the findings from both policies. The Oxford border closure data does not detail the target country for partial border closures. Given this data limitation, the manuscript could acknowledge that it is harder to evaluate the impact of targeted border closures, as they targeted different countries and these were not necessarily the countries where most cases were increasing. For example, Brazil singled out Venezuela for border closures in the early period of the pandemic. This is unlikely to be have been very effective. 

4. Figure 1: shouldn’t the y axis be “% of world population impacted”?

5. Methods. Please explain why linear regression model provide more conservative and directly comparable outcomes compared to exponential or logarithmic regression models? An explanation or a relevant citation must be provided.

Reviewers' comments:

Reviewer's Responses to Questions

**Comments to the Author**

1. If the authors have adequately addressed your comments raised in a previous round of review and you feel that this manuscript is now acceptable for publication, you may indicate that here to bypass the “Comments to the Author” section, enter your conflict of interest statement in the “Confidential to Editor” section, and submit your "Accept" recommendation.

Reviewer #3: (No Response)

2. Does this manuscript meet PLOS Global Public Health’s publication criteria? Is the manuscript technically sound, and do the data support the conclusions? The manuscript must describe methodologically and ethically rigorous research with conclusions that are appropriately drawn based on the data presented.

Reviewer #3: Yes

3. Has the statistical analysis been performed appropriately and rigorously?

Reviewer #3: Yes

4. Have the authors made all data underlying the findings in their manuscript fully available (please refer to the Data Availability Statement at the start of the manuscript PDF file)?

Reviewer #3: Yes

5. Is the manuscript presented in an intelligible fashion and written in standard English?

Reviewer #3: Yes

6. Review Comments to the Author

Reviewer #3: Thank you for the opportunity to review this manuscript of high-importance. This manuscript evaluated the impact of border-closure of 166 countries’ using multiple quasi-experimental methods. The results of the study indicate that total border closures had some effect on reducing COVID-19 transmission, and I agree with the recommendations that the authors have made. This manuscript is scientifically sound and well-written, yet there are some very minor points to address in order to strengthen the manuscript. I hope the authors well in their future research.

General points

Some of the statements made by the authors in the Introduction section imply that results from mathematical models are somewhat limited in evaluating the impact of border closures. While I agree real-world data provides better evidence for policy making, mathematical models are also highly valuable during the early stages of a public health event (especially when real-world data are scarce). This should be acknowledged. 

Figure 1: shouldn’t the y axis be “% of world population impacted”?

Line 131: Why does the linear regression model provide more conservative and directly comparable outcomes compared to exponential or logarithmic regression models? An explanation or a relevant citation must be provided.

7. PLOS authors have the option to publish the peer review history of their article (what does this mean?). If published, this will include your full peer review and any attached files.

**Do you want your identity to be public for this peer review?** For information about this choice, including consent withdrawal, please see our Privacy Policy.

Reviewer #3: No

---

## [Editor Report · Decision Letter 2]

23 Jan 2023

Quasi-experimental evaluation of national border closures on COVID-19 transmission

PGPH-D-21-01124R2

Dear Dr. Hoffman,

We are pleased to inform you that your manuscript 'Quasi-experimental evaluation of national border closures on COVID-19 transmission' has been provisionally accepted for publication in PLOS Global Public Health.

Best regards,

Lorena G Barberia

Section Editor
